# Cross-cultural adaptation and psychometric validation of a Spanish version of the Maryland Assessment of Recovery Scale (MARS-12)

**Nekane Balluerka**[1], **Arantxa Gorostiaga**[1], **Hernán María Sampietro**[2,3], **Ana González-Pinto**[1,4], **Jone Aliri**[1] *

1 Department of Clinical and Health Psychology and Research Methods, Faculty of Psychology, University of the Basque Country UPV/EHU, Donostia-San Sebastián, Spain, 2 Department of Social Psychology and Quantitative Psychology, Faculty of Psychology, University of Barcelona, Barcelona, Spain, 3 ActivaMent Catalonia Association, Barcelona, Spain, 4 Bioaraba, Department Psychiatry, Hospital Universitario de Alava, UPV/EHU, CIBERSAM, Vitoria-Gasteiz, Spain

* jone.aliri@ehu.eus

**Data Availability Statement:** All data files are available from the OSF Data repository (DOI: 10. 17605/OSF.IO/8U5YF).

## Abstract

The aim of this study was to adapt and validate a Spanish version of the Maryland Assessment of Recovery Scale (MARS-12). It was carried out in strict accordance with internationally recognized guidelines for test adaptation. A preliminary Spanish version of the MARS-12 was first produced through a standardized translation/back-translation process, ensuring semantic, linguistic, and contextual equivalence with respect to the original scale. Its psychometric properties were then examined in a sample of 325 people with serious mental illness recruited from six different provinces in the Basque Country (northern Spain) and Catalonia (north-eastern Spain). They were users of a total of 20 community rehabilitation and psychiatry services. Confirmatory factor analysis supported a unidimensional structure, consistent with the original scale. Scores on the MARS-12 were positively correlated (.83) with scores on the Questionnaire about the Process of Recovery, supporting convergent validity, while validity evidence based on relationships with other variables was provided by positive correlations between MARS-12 scores and scores on the Dispositional Hope Scale (.82) and on the three dimensions of the Multidimensional Scale of Perceived Social Support (range .30 to .41). Reliability of MARS-12 scores was high (McDonald's ω = .97), as was temporal stability across a one-week interval (.89). The Spanish version of the MARS-12 is a valid and reliable scale that may be used by mental health professionals to assess recovery among Spanish people with serious mental illness.

## Introduction

Through its recent QualityRights initiative, the WHO is seeking to promote a person-centered and rights-based approach to community mental health [1]. From this perspective, the primary goal of mental health policy should be to promote the wellbeing of people with mental

**Funding:** This study was supported by grant PID2019-109887GB-I00, funded by the Spanish Ministry of Science and Innovation (ref. MCIN/AEI/ 10.13039/501100011033); and by a grant from the Basque Government (ref. IT1493-22). The funders had no role in study design, data collection and analysis, decision to publish, or preparation of the manuscript.

**Competing interests:** The authors have declared that no competing interests exist.

illness enabling them to lead a satisfactory life. This goal is also what defines the recovery-based approach to mental health care [2]. The recovery philosophy began to take shape in the 1980s in the context of deinstitutionalization, promoted by the movement of users and survivors who advocated for services to go beyond the reduction of symptoms and to promote life in the community [3]. These ideas were rooted in the Civil Rights movement of the 1960s and 70s in the USA [4], which criticized the traditional psychiatry approach.

A central tenet of the recovery-oriented perspective is that recovery does not require a cure or the cessation of psychiatric symptoms and functional impairments. Rather, it is about a person regaining a sense of self and a meaningful life that reflects their values and expectations, and which enables them to engage with society [5]. Accordingly, recovery is a process where the emphasis is placed on acquiring or strengthening a connection with the community, hope for the future, a positive identity, meaning and purpose in life, and empowerment [6].

The recovery-oriented approach is not only aligned with the World Health Organization's perspective on mental health [1,7], but also with the United Nations Convention on the rights of persons with disabilities [8]. Nowadays, both institutions are promoting changes in their Member States to adjust state and regional legislation and norms to the framework of recovery-oriented and rights-based care [9].

This paradigm shift in mental health services began in the USA, UK and other English-speaking countries almost three decades ago [3], and was followed by several countries in northern Europe [10]. More recently, Spain [11], some countries in Latin America [12], and the global south [13] are following the same path. In the case of Spain, where the autonomous regions manage public health, several of them are also now seeking to align their mental health care policies more closely with the recovery-oriented approach, notably Catalonia [14], the Basque Country [15], and Navarre [16].

Despite the progress that has made in implementing the recovery model, Leonhardt et al. [17] concluded that although research into recovery from serious mental illness has grown significantly in recent decades, there remain several areas in need of further clarification. One of these concerns how recovery is measured, an issue that is crucial to the delivery of recovery-focused mental health services. In this sense, a recent scoping review on recovery planning tools, a type of resource that facilitates and promotes self-determination and self-management of the recovery process, found that although these types of tools are not oriented towards clinical recovery, the most usual variable evaluated in this area was the remission of symptoms and this would be especially marked in non-English speaking countries in which 43% of the studies only evaluated symptoms improvement [18].

In one of the most recent systematic reviews of instruments for measuring recovery, Penas et al. [19] highlighted the characteristics and quality of the Maryland Assessment of Recovery in People with Serious Mental Illness (MARS; [20]). They also noted that only two instruments for measuring recovery have so far been adapted for the Spanish-speaking population: The Recovery Assessment Scale (RAS; [21]), adapted by Zalazar et al. [22]; and the Stages of Recovery Instrument (STORI; [23]), adapted and validated by Lemos-Giráldez et al. [24] in Spanish individuals with a schizophrenia-spectrum disorder. More recently, Goodman-Casanova et al. [25] have published a Spanish adaptation of the 15-item Questionnaire about the Process of Recovery (QPR; [26]), while Saavedra et al. [27] have done the same with the shortened version of the Recovery Assessment Scale (RAS-24; [28]).

The limited availability of valid and reliable instruments for measuring recovery in Spanish people diagnosed of serious mental illness is a barrier to the delivery of recovery-focused services in our country. Consequently, our aim in this study was to adapt and validate for use in the Spanish population the MARS-12, a short form of the MARS [20] that was validated by Deborah Medoff (one of the authors of the original 25-item scale) and which is accessible in

Optum. The MARS fulfills the quality criteria established in the main systematic reviews of instruments for measuring recovery. In addition, it is based on a definition of recovery that is widely accepted by service users, namely that proposed by the SAMHSA (Substance Abuse and Mental Health Services Administration) in the USA: "Mental health recovery is a journey of healing and transformation enabling a person with a mental health problem to live a meaningful life in a community of his or her choice while striving to achieve his or her full potential" [29]. Our rationale for focusing specifically on the short form of the MARS (the MARS-12) was to prioritize ease of administration, thereby increasing the likelihood of the scale being used by service providers who are seeking to promote a more recovery-focused approach to mental health care.

The adaptation of the MARS-12 was carried out in accordance with the Standards for Educational and Psychological Testing [30] and the International Test Commission Guidelines [31].

The aim of this study was to examine the factor structure of the Spanish version of the MARS-12 and to evaluate the psychometric properties of the scale structure that showed the best fit to the data. This included assessment of internal consistency, convergent validity, and validity based on relationships with other variables. Previously, the translation and cultural adaptation of the MARS-12 were carried out following a standardized process aimed at ensuring semantic, linguistic, and contextual equivalence between the Spanish version and the original scale. The description of this process is provided in the Supplementary Material 1 (https://osf.io/8u5yf/?view_only=1357032b555047f8bb2e3574dbd70cd4).

## Materials and methods

### Participants

Participants were recruited through convenience sampling of three sources: associations for families of people with serious mental illness in the Basque Country (northern Spain), psychiatric services in the same region, and community rehabilitation services in the region of Catalonia (north-eastern Spain). The final sample comprised participants from six different provinces and who were users of a total of 20 community rehabilitation and psychiatry services. The minimum sample size required to obtain stable results in the confirmatory factor analysis (n = 100) was calculated based on the number of factors and indicators in the instrument being adapted, and by taking in account the conclusions of Wolf et al.'s [32] simulation study. We also determined that in order to approximate a representative sample of the population with serious mental illness, a total of 320 participants would be needed for an alpha of 5% and a margin of error of 2%.

The inclusion criteria were: age 18 or over and diagnosed, according to the criteria of the International Statistical Classification of Diseases and Related Health Problems (ICD-10; [33]), with one or more of the following disorders for at least two years: schizophrenia (F20), schizotypal disorder (F21), delusional disorder (F22), induced delusional disorder (F24), schizoaffective disorders (F25), other nonorganic psychotic disorders (F28), unspecified nonorganic psychosis (F29), bipolar affective disorders (F31), severe depressive episode with psychotic symptoms (F32.3) or recurrent depressive disorder (F33). Participants also had to be users of mental health services, to have a good command of Spanish, and to be able to complete the instruments alone or with the help of their key worker.

The final sample comprised 325 individuals (44% female) aged between 19 and 83 years (M = 47.82; SD = 10.51). The sociodemographic characteristics of participants are shown in Table 1.

All these participants completed the Spanish version of the MARS-12. To examine convergent validity and to obtain validity evidence based on relationships with other variables, two

**Table 1. Sociodemographic characteristics of the sample.**

|  | *n* | *%* |
|---|---|---|
| Civil status |  |  |
| Married or civil partner | 84 | 26.1 |
| Single | 167 | 51.9 |
| Divorced | 62 | 19.3 |
| Separated | 1 | 0.3 |
| Widowed | 8 | 2.5 |
| Current living situation |  |  |
| With family of origin (father/mother) | 121 | 37.9 |
| With own family (partner/children) | 92 | 28.9 |
| Alone | 59 | 18.5 |
| Shared apartment | 27 | 8.5 |
| Care facility | 7 | 2.2 |
| Other | 13 | 4.1 |
| Level of education |  |  |
| No formal education | 8 | 2.5 |
| Elementary | 106 | 33.0 |
| High school | 133 | 41.4 |
| University | 74 | 23.1 |

sub-samples also responded to additional instruments. One sub-sample, comprising 93 participants, completed the 15-item Spanish version of the Questionnaire about the Process of Recovery [25]. The second sub-sample, consisting of 230 participants, completed the Spanish versions of the Dispositional Hope Scale [34] and the Multidimensional Scale of Perceived Social Support [35].

## Instruments

**Spanish version of the MARS-12.** The translation and cultural adaptation of the MARS-12 was developed following the process described in the Supplementary Material 1 (https://osf.io/8u5yf/?view_only=459029f699fd4c2d806ffc551e4b0b78). As a result of such process, we obtained the version of the instrument validated in this study. It is a 12-item instrument designed to assess recovery in people with serious mental illness. Items are rated using a 5-point Likert-type scale (from 1 = Disagree strongly to 5 = Agree strongly), and hence the total score ranges between 12 and 60. Higher scores indicate a higher degree of perceived recovery. The items of the final Spanish version of the MARS-12, along with the original English items, are presented in the Supplementary Material 2 (https://osf.io/8u5yf/?view_only=1357032b555047f8bb2e3574dbd70cd4).

**Spanish version of the 15-item Questionnaire about the Process of Recovery (QPR; [25]; original scale [26]).** The QPR is a self-report instrument designed to ask people about aspects of recovery that are meaningful to them. Each of its 15 items is rated on a 5-point Likert-type scale (from 1 = Disagree strongly to 5 = Agree strongly), and hence the total score ranges from 15 to 75, with higher scores indicating greater perceived recovery. An example of an item on this scale is: "*I feel that my life has a purpose*". The Spanish version has shown good internal consistency [25]. The value of McDonald's ω in the present sample was .96.

**Spanish version of the Dispositional Hope Scale (DHS; [34]; original scale [36]).** The DHS is a self-report instrument that assesses people's ability to identify pathways and strategies for achieving their goals, and their motivation in seeking them. It comprises 12 items, each

rated on a 4-point Likert-type scale (from 1 = Definitely false to 4 = Definitely true). The total score ranges from 4 to 32 (four items are fillers and are not considered when computing the total score), with higher scores indicating greater dispositional hope. An example of an item on this scale is: "*My past experiences have prepared me well for my future*". The Spanish version has shown good internal consistency [34]. The value of McDonald's ω in the present sample was .56.

**Spanish version of the Multidimensional Scale of Perceived Social Support (MSPSS),** adapted by Ruiz Jiménez et al. [35] in a sample of people with serious mental illness (original scale [37]). The MSPSS is a self-report instrument comprising 12 items, each rated on a 7-point Likert-type scale (from 1 = Very strongly disagree to 7 = Very strongly agree). The items relate to three dimensions of perceived social support (4 items per dimension), namely that received from friends, family, and significant others. Example items for each of these dimensions are, respectively: "*My friends really try to help me*"; "*I get the emotional help and support I need from my family*"; and "*There is a special person who is around when I am in need*". The score for each dimension ranges from 1 to 28. The Spanish version showed good internal consistency in a sample of people with serious mental illness [35]. In the present sample, the value of McDonald's ω was .97, .96, and .83 for the dimensions friends, family, and significant others, respectively.

**Questionnaire about sociodemographic variables and mental health diagnosis.**  This was an ad hoc questionnaire designed to collect data from participants regarding age, gender, civil status, current living situation, and level of education.

**Procedure.**  Data were collected between January 24[th] and October 7[th] 2022. All participating service users and their families or legal guardians were informed about the nature and purpose of the study, it being made clear that participation was voluntary and that all data would remain confidential. For service users recruited through family associations, the instruments were administered by two psychologists of the research team, with support from users' key workers. In the case of users recruited through psychiatry services in the Basque Country, data were collected by two psychiatrists employed within these services. Data from users of community rehabilitation services in Catalonia were collected by a psychologist of the research team.

In all cases, participants first completed the questionnaire about sociodemographic variables, followed by the MARS-12, the QPR, the DHS, and the MSPSS.

The present study was conducted in accordance with the ethical principles of the World Medical Association and the latest version of the Declaration of Helsinki regarding research involving humans. The study was approved by the Bioethics Committee of the University of Barcelona on 29 November 2021. Responses to all the instruments used were given anonymously and written informed consent of the participants was obtained after the nature of the procedures had been fully explained.

## Data analysis

For the validation of the Spanish version of the MARS-12 we analyzed the following aspects:

a. *Items*: This involved calculation of descriptive statistics (mean, standard deviation, skewness, and kurtosis) and the corrected index of homogeneity.

b. *Dimensionality and construct validity*: Here we conducted a confirmatory factor analysis using the weighted least squares mean and variance adjusted (WLSMV) estimation method. Model fit was assessed by calculating the chi-squared statistic, the Tucker-Lewis index (TLI), the comparative fit index (CFI), and the root mean square error of approximation (RMSEA).

c. *Reliability*: Internal consistency was assessed through McDonald's ω coefficient. Temporal stability was estimated by calculating Pearson's correlation coefficient between scores obtained at two time points (one week apart) in a sub-sample of 66 service users.

d. *Convergent validity*: This was assessed by computing the Pearson correlation coefficient between scores obtained on the MARS-12 and scores on the QPR.

e. *Validity based on relationships with other variables*: This was assessed by computing the Pearson correlation coefficient between scores obtained on the MARS-12 and scores on two other measures: the DHS and each dimension of the MSPSS.

Data analyses were performed using Mplus and SPSS 28. Multiple imputation was used for missing values.

## Results

### Analysis of Scale Items

Table 2 shows descriptive statistics for scores on the Spanish version of the MARS-12. Mean scores on all items were above the mid-point of the Likert scale, and standard deviations were close to 1. Means ranged from 2.67 (item 8) to 3.58 (item 12), and standard deviations from 1.06 (item 2) to 1.36 (item 7). The distribution of three items (5, 11, and 12) showed significant negative skewness. All items showed a platycurtic distribution. Finally, the corrected index of item homogeneity was .63 or higher.

### Dimensionality and construct validity

In the confirmatory factor analysis carried out to determine whether the Spanish version of the MARS-12 had a unidimensional factor structure, all but one of the indices indicated a good fit of this model to the data. The exception was the RMSEA, the value of which was slightly above the threshold for acceptable fit ($\chi^2$ (54) = 219.863; $p < .001$; CFI = .98; TLI = .98; RMSEA = .10).

It can be seen in Table 3 that standardized factor loadings for items of the Spanish version of the MARS-12 were all .68 or higher.

**Table 2. Means, standard deviations, skewness and kurtosis indices, and item homogeneity for scores on the Spanish version of the MARS-12.**

| Item | M | SD | Skewness | Kurtosis | Homogeneity index |
|---|---|---|---|---|---|
| MARS1 | 2.82 | 1.23 | 0.08 | -0.89** | .79 |
| MARS2 | 3.01 | 1.06 | -0.10 | -0.58* | .69 |
| MARS3 | 2.89 | 1.17 | 0.00 | -0.81** | .77 |
| MARS4 | 3.03 | 1.29 | -0.03 | -1.09** | .72 |
| MARS5 | 3.39 | 1.10 | -0.38** | -0.56* | .81 |
| MARS6 | 3.15 | 1.25 | -0.20 | -0.97** | .65 |
| MARS7 | 3.07 | 1.36 | -0.12 | -1.19** | .68 |
| MARS8 | 2.67 | 1.33 | 0.25 | -1.10** | .79 |
| MARS9 | 3.05 | 1.28 | -0.05 | -1.04** | .82 |
| MARS10 | 3.02 | 1.28 | -0.05 | -1.07** | .86 |
| MARS11 | 3.41 | 1.15 | -0.37** | -0.69* | .77 |
| MARS12 | 3.58 | 1.15 | -0.49** | -0.63* | .63 |

\* $p < .05$

\*\* $p < .01$

**Table 3. Standardized factor loadings, unstandardized factor loadings, and standard errors for items of the Spanish version of the MARS-12.**

| Item | Standardized factor loadings | Unstandardized factor loadings | Standard errors |
|---|---|---|---|
| MARS1 | .857 | 1.000 | .000 |
| MARS2 | .749 | 0.874 | .030 |
| MARS3 | .823 | 0.961 | .024 |
| MARS4 | .770 | 0.899 | .027 |
| MARS5 | .869 | 1.015 | .022 |
| MARS6 | .703 | 0.821 | .034 |
| MARS7 | .743 | 0.868 | .031 |
| MARS8 | .833 | 0.973 | .025 |
| MARS9 | .888 | 1.037 | .021 |
| MARS10 | .921 | 1.076 | .021 |
| MARS11 | .847 | 0.988 | .025 |
| MARS12 | .685 | 0.800 | .034 |

### Convergent validity

The correlation between scores on the Spanish version of the MARS-12 and those on the QPR was .83, indicating adequate convergent validity.

### Validity based on relationships with other variables

The correlation between scores on the Spanish version of the MARS-12 and those on the DHS was .82, indicating, as expected, that people who perceive a higher degree of recovery also report higher levels of dispositional hope. The correlation coefficients between scores on the MARS-12 and those on the three dimensions of the MSPSS (corresponding, respectively, to perceived social support from friends, family, and significant others) were .41, .36, and .30. Although these correlations are of moderate magnitude, they suggest, as expected, that people who perceive a higher degree of recovery also perceive higher levels of social support from friends, family, and significant others.

### Reliability

McDonald's ω coefficient for the Spanish version of the MARS-12 was .97, indicating that the scale has high internal consistency. Regarding temporal stability, the correlation between scores obtained on the MARS-12 at the two time points was .89, which may be interpreted as indicating high temporal stability.

## Discussion

The effective delivery of recovery-focused mental health care requires appropriate and validated instruments for assessing the outcomes of psychiatric and psychosocial interventions. To this end, the present study sought to cross-culturally adapt and validate a Spanish version of the MARS-12, a short scale that evaluates recovery from serious mental illness from the service user's perspective.

The scale adaptation was conducted in accordance with the Standards for Educational and Psychological Testing [30] and the International Test Commission Guidelines [31]. The results obtained in the translation and cultural adaptation of the instrument provided confirmation of semantic, linguistic, and contextual equivalence between the Spanish version of the MARS-12 and the original scale, thus providing evidence of content validity. After translating and adapting the Spanish version of the MARS-12, we examined its psychometric properties in a large

sample of people with serious mental illness. Item analysis indicated adequate scale properties. Only three items showed significant skewness, which in each case was negative and of small magnitude, indicating that scores are adequately distributed. Regarding kurtosis, although all items showed a platykurtic distribution, the kurtosis indices were low (below an absolute value of 1.5 in all cases).

Consistent with the factor structure obtained when developing the original 25-item MARS [20], the Spanish version of the MARS-12 showed a unidimensional structure with all items clearly loading on a single factor. Evidence of convergent validity was provided by a positive correlation of high magnitude between scores on the Spanish version of the MARS-12 and scores on the QPR, which measures the same construct. Regarding validity evidence based on relationships with other variables, the correlations observed between scores on the Spanish version of the MARS-12 and scores on the DHS and on the three dimensions of the MSPSS indicate that people who perceive a higher level of recovery also report greater dispositional hope and feel more supported by friends, family, and significant others. These relationships are consistent with what would expect from the perspective of the recovery-oriented framework [6].

With respect to internal consistency, the coefficients obtained for the Spanish version of the MARS-12 were similar to those reported for the original 25-item scale [20] and higher than those obtained in a subsequent validation study [38]. As regards test-retest reliability, the coefficient obtained was again similar to that reported by Drapalski et al. [20] and higher than that obtained in their later study [38], supporting good temporal stability for the Spanish version of the MARS-12. It should be noted that in 2013 Shanks et al. [39] published a systematic review of personal recovery measures, observing that in the original MARS study [20] only 4 psychometric properties of the scale had been evaluated. The subsequent research made it possible to extent such validation study [40]. The present study has taken into account the advantages of both previous studies and has filled the gap regarding the absence of a study that exhaustively evaluated the psychometric properties of the short version of the MARS. (The final Spanish version of the MARS-12, including the instructions and the response options for completing it are provided in the Supplementary Material 3 https://osf.io/8u5yf/?view_only=1357032b555047f8bb2e3574dbd70cd4).

In summary, the Spanish version of the MARS-12 appears to be a valid and reliable scale that could be used by mental health professionals to assess recovery among Spanish people with serious mental illness. Given that adequate assessment of recovery is a key component in the delivery of recovery-focused mental health care, and bearing in mind the lack of validated recovery assessment instruments in our sociocultural setting, the availability of a Spanish version of the MARS-12 makes an important contribution to the field. We believe that the scale's properties make it suitable for use within the context of recovery programs that are currently being implemented and evaluated in Spain [40]. Furthermore, it can drive the development of new mental health strategic plans in other autonomous regions of this country. From a professional's point of view, the main contribution that the MARS-12 can make is to facilitate the implementation of evidence-based interventions aimed at promoting recovery. From the perspective of users and families, the scale could encourage interventions to go beyond clinical objectives, which are focused on controlling symptoms and functionality, thus responding to a historical request from organisations that represent them [41] From an international perspective, this new version of the MARS-12, adapted and validated in Spain, also provides a platform for further adaptations within the Spanish-speaking world, potentially extending the applicability of this tool to the population of over 20 countries.

The main limitation of this study is that participants were recruited by convenience sampling within specific regions of Spain. That said, the number of participants was sufficient to approximate a representative sample of service users with serious mental illness.

Notwithstanding this limitation, the study also has two key strengths. One is that the Spanish version of the MARS-12 was developed in strict accordance with internationally recognized guidelines for test adaptation. In addition, its psychometric properties were examined in a sizable sample of the target population (i.e., Spanish adults with serious mental illness).

The results obtained in this study suggest that the Spanish version of the MARS-12 is a valid and reliable instrument for assessing recovery among Spanish people with serious mental illness. In terms of practical implications, the availability of this new tool, which is both short and easy to administer, offers professionals a ready way of assessing recovery, and accordingly it should make a positive contribution to the delivery of recovery-focused mental health care in Spain. Finally, from an integrative perspective, having a scale that allows evaluating personal recovery could facilitate the incorporation into the portfolio of mental health services some resources of the WHO Quality Rights initiative aimed at promoting self-determination and self-management of the recovery process, such as advanced care planning [42] and recovery planning tools [43].

## Author Contributions

**Conceptualization:** Nekane Balluerka.

**Data curation:** Hernán María Sampietro.

**Formal analysis:** Arantxa Gorostiaga, Jone Aliri.

**Funding acquisition:** Nekane Balluerka.

**Investigation:** Hernán María Sampietro, Ana González-Pinto, Jone Aliri.

**Methodology:** Arantxa Gorostiaga.

**Project administration:** Arantxa Gorostiaga.

**Resources:** Hernán María Sampietro, Ana González-Pinto.

**Software:** Jone Aliri.

**Supervision:** Nekane Balluerka.

**Validation:** Nekane Balluerka.

**Visualization:** Arantxa Gorostiaga.

**Writing – original draft:** Nekane Balluerka.

**Writing – review & editing:** Ana González-Pinto, Jone Aliri.

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
