## [Decision Letter · Decision Letter 0]

20 Dec 2023

PONE-D-23-29843Cross-cultural Adaptation and Psychometric Validation of a Spanish Version of the Maryland Assessment of Recovery Scale (MARS-12)PLOS ONE

Dear Dr. Aliri,

Thank you for submitting your manuscript to PLOS ONE. After careful consideration, we feel that it has merit but does not fully meet PLOS ONE’s publication criteria as it currently stands. Therefore, we invite you to submit a revised version of the manuscript that addresses the points raised during the review process.

Please submit your revised manuscript by Feb 03 2024 11:59PM. If you will need more time than this to complete your revisions, please reply to this message or contact the journal office at plosone@plos.org. Please include the following items when submitting your revised manuscript:A rebuttal letter that responds to each point raised by the academic editor and reviewer(s). You should upload this letter as a separate file labeled 'Response to Reviewers'.A marked-up copy of your manuscript that highlights changes made to the original version. You should upload this as a separate file labeled 'Revised Manuscript with Track Changes'.An unmarked version of your revised paper without tracked changes. You should upload this as a separate file labeled 'Manuscript'.

We look forward to receiving your revised manuscript.

Kind regards,

Maria José Nogueira, Ph.D.

Academic Editor

PLOS ONE

Journal Requirements:

Did you know that depositing data in a repository is associated with up to a 25% citation advantage (https://doi.org/10.1371/journal.pone.0230416)? If you’ve not already done so, consider depositing your raw data in a repository to ensure your work is read, appreciated and cited by the largest possible audience. You’ll also earn an Accessible Data icon on your published paper if you deposit your data in any participating repository (https://plos.org/open-science/open-data/#accessible-data).

4. We note that you have referenced (ie. Optum. Maryland Assessment of Recovery Scale — MARS-12 [Internet]) which has currently not yet been accepted for publication. Please remove this from your References and amend this to state in the body of your manuscript: (ie “Bewick et al. [Unpublished]”) as detailed online in our guide for authors

**Additional Editor Comments:**

The study presents original research.

The authors should clearly highlight the new contributions that MARS-12 adds to the practice.

Revise the state of the art (with more recent evidence), discussion, and conclusions as sugested.

Reviewers' comments:

Reviewer's Responses to Questions

**Comments to the Author**

1. Is the manuscript technically sound, and do the data support the conclusions?

Reviewer #1: Yes

Reviewer #2: Partly

2. Has the statistical analysis been performed appropriately and rigorously? 

Reviewer #1: Yes

Reviewer #2: Yes

3. Have the authors made all data underlying the findings in their manuscript fully available?

Reviewer #1: Yes

Reviewer #2: Yes

4. Is the manuscript presented in an intelligible fashion and written in standard English?

Reviewer #1: Yes

Reviewer #2: Yes

5. Review Comments to the Author

Reviewer #1: Very interesting topic for the practice of Mental Health and Psychiatric Nursing.

In general, it is well structured, complying with the guidelines for the adoption and validation of a scale

The research took into account credible scientific databases and other complementary literature.

Reviewer #2: Thanks you for your article titled "Cross-cultural Adaptation and Psychometric Validation of a Spanish Version of the Maryland Assessment of Recovery Scale (MARS-12)." Overall, this study has been conducted appropriately, meeting all the criteria required for research of this nature.

The article is well-prepared for publication, with a clear structure, comprehensive methodology, and relevant results. The topic is timely and significant. However, it is crucial to highlight the benefits of this instrument for care teams, users, and their families. Subsequently, strengthening the discussion of the data and presenting the study's conclusion would be beneficial.

My main concern relates to the literature review, with 70% of the documents used being over 5 years old.

Specific Comments:

Abstract:

The abstract should clearly outline the method used in this study, and it would be helpful to specify the contexts to which the 325 participants belong.

Introduction:

Consider expanding the literature review in the introduction, focusing on current research on self-care management and self-monitoring Clarify the benefits of the instrument for professionals, users, and families.

Materials and Methods:

The materials and methods section appears to be adequate and well-explored, according to your feedback.

Results:

Confirm that the results are presented clearly and concisely.

Discussions (lines 275-312):

The discussion section deserves further exploration. Include references from authoritative authors regarding the scale's evaluated items and their benefits.

Make comparisons with recent studies on the topic, using more up-to-date bibliography.

Evaluate whether the current discussion is sufficient for such a relevant and pertinent theme and if it adequately represents the work done in this study.

Conclusion:

Clearly identify the conclusions drawn from the study.

References:

Note that references, on the whole, are over 5 years old and suggest improvements by incorporating more recent sources.

6. PLOS authors have the option to publish the peer review history of their article (what does this mean?). If published, this will include your full peer review and any attached files.

Reviewer #1: No

Reviewer #2: No

---

## [Author Response · Author response to Decision Letter 0]

24 Jan 2024

First of all, we would like to express our gratitude for the reviews received. We believe that the reviewers´ suggestions have allowed us to improve the quality of our manuscript. All changes in the revised manuscript are marked with track changes in the paper we have submitted via the online submission system. In the next few lines, we respond to the Journal requirements, to additional editor’s comments and the reviewers’ comments one by one.

RESPONSE TO JOURNAL REQUIREMENTS:

We have ensured that the manuscript meets PLOS ONE’s style requirements, including those for file naming. 

Did you know that depositing data in a repository is associated with up to a 25% citation advantage (https://doi.org/10.1371/journal.pone.0230416)? If you’ve not already done so, consider depositing your raw data in a repository to ensure your work is read, appreciated and cited by the largest possible audience. You’ll also earn an Accessible Data icon on your published paper if you deposit your data in any participating repository (https://plos.org/open-science/open-data/#accessible-data).

We have deposited data in the OSF: DOI 10.17605/OSF.IO/8U5YF

We have deposited the data in the OSF: DOI 10.17605/OSF.IO/8U5YF

4. We note that you have referenced (ie. Optum. Maryland Assessment of Recovery Scale — MARS-12 [Internet]) which has currently not yet been accepted for publication. Please remove this from your References and amend this to state in the body of your manuscript: (ie “Bewick et al. [Unpublished]”) as detailed online in our guide for authors.

We have removed the unpublished reference and amend it in the body of the manuscript.

RESPONSE TO ADDITIONAL EDITOR COMMENTS:

• The study presents original research.

• The authors should clearly highlight the new contributions that MARS-12 adds to the practice.

We have clearly highlighted the contributions that MARS-12 adds to the practice (pages 17 (lines 356-363) and 18 (lines 381-385) in the track changes version). 

• Revise the state of the art (with more recent evidence), discussion, and conclusions as sugested.

We have added new ideas based on recent evidence both in the introduction and in the discussion. In fact, we have added 17 new references. 

RESPONSE TO REVIEWER #1: 

Very interesting topic for the practice of Mental Health and Psychiatric Nursing.

In general, it is well structured, complying with the guidelines for the adoption and validation of a scale.

The research took into account credible scientific databases and other complementary literature.

We are very grateful for your kind words.

RESPONSE TO REVIEWER #2: 

Thanks you for your article titled "Cross-cultural Adaptation and Psychometric Validation of a Spanish Version of the Maryland Assessment of Recovery Scale (MARS-12)." Overall, this study has been conducted appropriately, meeting all the criteria required for research of this nature.

The article is well-prepared for publication, with a clear structure, comprehensive methodology, and relevant results. The topic is timely and significant. However, it is crucial to highlight the benefits of this instrument for care teams, users, and their families. Subsequently, strengthening the discussion of the data and presenting the study's conclusion would be beneficial.

My main concern relates to the literature review, with 70% of the documents used being over 5 years old.

Specific Comments:

Abstract:

• The abstract should clearly outline the method used in this study, and it would be helpful to specify the contexts to which the 325 participants belong. 

Following the reviewer’s suggestion, he have included in the abstract information on the method used in the study and the context to which the participants belong.

Introduction:

• Consider expanding the literature review in the introduction, focusing on current research on self-care management and self-monitoring Clarify the benefits of the instrument for professionals, users, and families.

We have expanded the literature review in the introduction, focusing on current research on the recovery-based approach to mental health care which emphasizes self-care management and self-monitoring. We have also added a last paragraph highlighting the practical implications of this study and adding some ideas about the benefits of the instrument to promote self-determination and self-management in the recovery process.

We have added new ideas and 17 recent references in order to reflect the current conceptualization of the recovery-oriented perspective. 

Furthermore, we have clarified, in the discussion (page 16) the benefits of the instrument for professionals, users and families. 

Materials and Methods:

• The materials and methods section appears to be adequate and well-explored, according to your feedback.

Results:

• Confirm that the results are presented clearly and concisely.

Discussions (lines 275-312):

• The discussion section deserves further exploration. Include references from authoritative authors regarding the scale's evaluated items and their benefits.

We have added a reference to a systematic review focused on personal recovery measures and we have used it as a basis for pointing out the advantages of the MARS-12 over previously developed long versions of the scale.

• Make comparisons with recent studies on the topic, using more up-to-date bibliography.

We have answered to this suggestion adding more recent studies on the topic in the introduction and in the discussion.

• Evaluate whether the current discussion is sufficient for such a relevant and pertinent theme and if it adequately represents the work done in this study.

Conclusion:

• Clearly identify the conclusions drawn from the study. 

In the last paragraph of the discussion of the original version of the manuscript some conclusions derived from the study were already summarized. However, following the suggestion of the reviewer we have added a last paragraph focusing mainly on the practical implications of this work so that they can be clearly identified.

References:

• Note that references, on the whole, are over 5 years old and suggest improvements by incorporating more recent sources.

We have incorporated 17 more recent references throughout the study and we have removed 7 older ones.

We look forward to hearing from you in due time regarding our submission and to respond to any further questions and comments you may have.

Sincerely,

Jone Aliri

---

## [Editor Report · Decision Letter 1]

26 Jan 2024

Cross-cultural Adaptation and Psychometric Validation of a Spanish Version of the Maryland Assessment of Recovery Scale (MARS-12)

PONE-D-23-29843R1

Dear Dr. Jone Aliri,

We’re pleased to inform you that your manuscript has been judged scientifically suitable for publication and will be formally accepted for publication once it meets all outstanding technical requirements.

Kind regards,

Maria José Nogueira, Ph.D.

Academic Editor

PLOS ONE

Additional Editor Comments (optional):

The authors carried out the revisions suggested by the reviewers and the Editor, which made the manuscript more robust and precise.

Therefore, I consider that the manuscript can be accepted for publication.
---

## [Editor Report · Acceptance letter]

14 Feb 2024

PONE-D-23-29843R1 

PLOS ONE

Dear Dr. Aliri, 

I'm pleased to inform you that your manuscript has been deemed suitable for publication in PLOS ONE. Congratulations! Your manuscript is now being handed over to our production team.

Kind regards, 

on behalf of

Professor Maria José Nogueira 

Academic Editor

PLOS ONE